# TWIK Complex Expression in Prostate Cancer: Insights into the Biological and Therapeutic Significances of Potassium Ion Channels in Clinical Cancer

**DOI:** 10.3390/biology14050569

**Published:** 2025-05-19

**Authors:** Abdulaziz Alfahed

**Affiliations:** Department of Medical Laboratory, College of Applied Medical Sciences, Prince Sattam bin Abdulaziz University, Al-Kharj 11942, Saudi Arabia; a.alfahed@psau.edu.sa

**Keywords:** prostate cancer, TWIK complex, pathological stage, androgen deprivation therapy, Gleason grade, ISUP prognostic grade group, aberrant ubiquitination, glucuronidation, ion channel blockers, chemotherapeutic drugs

## Abstract

The death rate from prostate cancer worldwide is still high; hence, there is a need to investigate the biological mechanisms of the disease in order to identify cancer cell vulnerabilities. This study investigates the roles of the potassium channel complex, called TWIK, in natural prostate cancer in actual patients. The rationale for studying the TWIK complex in prostate cancer is that many functions have been found for ion channels in cancer cells growing in experimental cultures, but the implications of these functions in natural cancers have not been studied in depth. This study investigates the roles of the TWIK complex in promoting prostate cancer growth. It also investigates the potential for using TWIK complex expression to predict whether prostate cancer patients would respond to certain types of anti-cancer drugs. The results show that, indeed, the TWIK complex supports the growth of cancer cells in real patients. They also show that the expression of the TWIK complex may be able to predict which anti-cancer drugs cancer patients may respond to. In addition, they reveal that cancers with high TWIK complex expression may have potential therapeutic targets.

## 1. Introduction

The need to relentlessly investigate the molecular pathogenesis of prostate cancer (PCa) cannot be overemphasised, inasmuch as the epidemiological profile of the disease remains poor. PCa is the fourth most common cancer worldwide, and the eighth most common cause of cancer-related deaths [1], in spite of the significant effort and resources that have been expended towards the understanding its molecular pathogenesis [2,3]. Currently, risk prediction using biochemical and clinicopathological parameters is the bedrock of PCa management [4,5]. However, current risk prediction schemes suffer from severe limitations [6,7,8], and it is difficult for urologists to predict which patients will develop more aggressive disease based on the current schemes [9,10]. The predictive and therapeutic significances of molecular alterations in PCa are not fully understood. Additional research is needed to investigate the molecular drivers of PCa aggressiveness and to identify biomarkers for diagnosis and treatment.

This study delves into the roles of ion channels in the pathogenesis and progression of clinical PCa, with a view to clarifying their clinical, molecular, and therapeutic significances in the disease. The activities of ion channels have been examined in in vitro and in vivo cancer studies. Through these studies, it has been determined that ion channels are active in the establishment of all eight hallmarks of cancer [11], such that cancer is now regarded as a channelopathy [12]. Furthermore, it has been demonstrated that targeting ion channels may be a veritable strategy for the treatment of cancer [13,14,15,16,17,18]. Therapeutic targeting of ion channels in cancer is an attractive treatment strategy [13,14] since ion channels appear to have ubiquitous functions in tumour cells in vitro [11,19,20]. However, these findings have yet to be translated into clinical use; studies validating the clinical usefulness of the findings of the in vitro studies are few [21,22,23,24,25,26].

Specifically, this study investigates the expression of the TWIK (tandem of P-domains in a weakly inward rectifying K+) subfamily of KCNK channels in PCa. KCNK is the K subfamily of the two-pore domain potassium (K+) channels [19,20]. The TWIK subfamily within the KCNK group consists of three members, including KCNK1 (TWIK-1), KCNK6 (TWIK-2), and KCNK7 [19,20]. KCNK genes encode two-pore-domain potassium ion transmembrane protein complexes that control the inflow of ions across biological membranes, thereby promoting the maintenance of resting membrane potential [27,28]. The maintenance of membrane potentials, in turn, is essential for many cellular functions, including metabolic regulations [29,30] apoptosis [31,32], chemoperception [33,34,35], maintenance of the blood–brain barrier [36,37], the maintenance of systemic and pulmonary blood pressure [38,39], and many others [27,40]. The maintenance of membrane potentials probably also underlies the oncogenic roles that have been described for members of the KCNK protein family in preclinical and limited clinical cancer studies [28,41,42,43,44,45].

In this study, a gene interactive or molecular network/complex approach to investigating the biological significances of ion channels in PCa was adopted, inasmuch as ion channels function as complexes or interactive molecular networks [46,47]. The aims of this study are to clarify whether the TWIK complex functions as an oncogene or a tumour suppressor gene in clinical PCa, and whether it contributes to PCa progression. This clarification is relevant to the therapeutic targeting of the complex for cancer treatment. The specific objectives are (i) to determine the relationship pattern between the TWIK complex and clinicopathological/molecular indices, including roles in tumour progression and associations with oncogenic signalling and onco-metabolism, and (ii) to investigate the potential therapeutic significances of the TWIK complex in PCa, including its relationship with androgen deprivation therapy, ion channel blockers, and conventional chemotherapeutic agents. The study hypothesis is that since the TWIK complex functions in the maintenance of basal cellular states and contributes to the hallmark of cancer in vitro, it can function as an oncogene in clinical cancer and may be associated with adverse clinicopathological, molecular, and prognostic indices of PCa. As an ion channel complex, its expression can be associated with gene expression programmes that specify responses to drugs of diverse mechanisms of action. The rationale for this study is the dearth of studies investigating the clinical relevance of the TWIK gene complex in PCa.

## 2. Materials and Methods

### 2.1. PCa Cohorts

This study retrospectively analysed the clinicopathological and genomic data of three PCa cohorts contained in the Genome Data Commons (GDC) and cBioPortal databases [48,49]: The Cancer Genome Atlas (TCGA) [10,48,49,50], Deutsches Krebsforschungszentrum (DKFZ) [50,51], and the MSKCC Prostate Oncogenome Project cohorts [10,52].

### 2.2. Study Approach

First, the subfamily network approach was adopted based on the structural similarities among its members, which suggest that they have similar functions and probably function as a complex. The interaction of the TWIK subfamily members as a molecular network was validated with protein–protein interaction (PPI) analysis on the STRING database (https://string-db.org/ accessed on 7 January 2025) [53]. A gene expression signature was utilised as a surrogate for TWIK complex activity, as previously demonstrated for tumour cell microvascular invasion activity [54]. The gene expression signature for the TWIK complex was generated using the geometric mean of the mRNA expression levels of *KCNK1*, *KCNK6*, and *KCNK7* [55,56]. Next, the gene expression signature was investigated in the TCGA and other cohorts for its relationship with clinicopathological, molecular, and therapeutic indices. Specifically, the TWIK complex signature was used as a phenotype (high vs. low) in gene set enrichment analysis (GSEA) and gene ontology enrichment analysis (GOEA) to determine the biological (including metabolic) and therapeutic significances of TWIK complex expression in the PCa cohorts.

### 2.3. Data Retrieval and Processing

The clinical and genomic data of interest were extracted from the TCGA, DFKZ, and MSKCC PCa cohorts using Linux-based scripts and codes, which were written in the Windows-based Ubuntu 20.04 environment. The genomic data comprised RNASeq, and masked segment data of 500, 95, and 150 PCa cases from the TCGA, DFKZ, and MSKCC cohorts, respectively. Data transformation and normalisation per cohort were accomplished using fractional ranking and the method described by Templeton [57], respectively. The transformed data from all three cohorts were merged into a combined cohort for assessment of the clinicopathological relevancies of the TWIK complex in PCa.

### 2.4. Gene Expression Signature Generation

To generate the TWIK complex signature, the expression levels of *KCNK1*, *KCNK6*, and *KCNK7* from the TCGA cohort were analysed using Cox regression to determine the direction of the correlations between overall survival and the individual gene expressions. Then, the TWIK complex signature was generated using the formula below:TWIK complex signature = geometric mean of expression values of the positively correlated genes/geometric mean of expression values of the negatively correlated genes(1)

### 2.5. Gene Set Enrichment Analysis

The biological associations of the TWIK gene signature were investigated using GSEA, which was performed using GSEA_4.3.3 software with the hallmark cancer signalling gene sets, as per GSEA requirements [58,59]. Differential enrichments of human metabolic gene set collections were also sought between TWIK-low and TWIK-high cases. These gene set collections were extracted from the human genome-scale metabolic model (Human-GEM) database (https://github.com/JonathanRob/GeneSetAnalysisMatlab/tree/master/gsc accessed on 3 January 2025) [60]. GSEA was also performed for gene sets with specific responses or perturbations to ion channel blockers and conventional chemotherapeutic agents (DSigDB, https://dsigdb.tanlab.org/DSigDBv1.0/ accessed on 4 February 2025) [61]. GSEA was validated by GOEA and Pathway Analysis in the Enrichr environment (https://maayanlab.cloud/Enrichr/enrich accessed on 14 January 2025) [62]. The phenotype and derivative gene set files were prepared in an Excel spreadsheet and converted into cls and gct files, respectively. Due to the unequal records of genes in the individual datasets (TCGA = 20,531; DFKZ = 20,881, and MSKCC = 26,477), the three PCa cohorts were individually investigated for the biological and predictive significance of the TWIK complex using GSEA. The process for GSEA on the three different datasets were performed as per MSigDB requirements. GSEA with leading edge analysis was first performed on one of the datasets using a collection of gene sets for a feature (such as oncogenic signalling). The core enrichment genes for the gene set collection were obtained and converted into a gene list. The gene list was then used as a single gene set in a subsequent GSEA on the other two datasets to obtain genes that were commonly enriched among the three data sets. The commonly enriched genes were then investigated for pathway and ontology terms enrichment in Enrichr (using adjusted *p* value of <0.05 as the threshold of enrichment). The enriched pathways and ontology terms were examined between the TWIK-high and TWIK-low PCa subsets to obtain differential enrichments, i.e., pathways and ontology terms that were unique to either subset. Most of the derived data were displayed with charts directly compiled from the Enrichr platform. The others were produced on the SRplot platform (https://www.bioinformatics.com.cn/en?keywords=basic_plot accessed on 7 January 2025) [63].

### 2.6. Statistical Analyses

The relationships of the TWIK complex signature with the clinicopathological, molecular, and genomic features of PCa were analysed using SPSS version 29. Fractional ranking and normalisation of continuous data were also accomplished using SPSS. A chi-squared (or Fisher’s) test was used to test for significant associations between categorical variables, while bivariate correlative analysis was utilised to examine the correlations between continuous variables. One-way ANOVA was used to measure the mean differences in continuous variables between discrete groups, while multivariate analysis was investigated with regression analyses. A *p* value of <0.05 was taken as the threshold for significant association or correlation. The Benjamini–Hochberg correction was applied for multiple testing using a false discovery rate (FDR) of 0.05. GSEA was performed with a default threshold nominal rate of 0.05 and an FDR of 0.25. Permutation number was maintained as 1000, while permutation type was set to “gene set”.

## 3. Results

### 3.1. PPI and Cox Regression Analyses Confirm the TWIK Subfamily as a Gene Network 

Gene network analysis on the STRING database showed a high probability that *KCNK1*, *KCNK6*, and *KCNK7* function as a gene network (average local clustering coefficient = 1; PPI enrichment *p*-value = 1.01 × 10^−9^). Cox regression analysis of the expressions of *KCNK1*, *KCNK6*, and *KCNK7* from the TCGA dataset showed that whilst *KCNK1* and *KCNK7* demonstrated a direct correlation with overall survival, *KCNK6* was inversely correlated with overall survival. The TWIK complex signature was generated based on this observed relationship between the TWIK subfamily genes and overall survival status.

### 3.2. TWIK Signature Exhibits Associations with Clinicopathological and Molecular Features

One-way ANOVA showed that the TWIK complex signature exhibited significant correlations with the clinicopathological indices, including the pathological tumour stage, pathological node stage, overall TNM stage, Gleason score, ISUP grade group score, response to androgen deprivation therapy, prostate-specific antigen levels, and fraction genome altered (FGA) (Appendix A). The chi-squared test confirmed the association of the TWIK complex signature with the above features (Figure 1 and Figure 2). Furthermore, the TWIK complex signature predicted overall and disease-free survivals (TCGA cohort) and time-to-biochemical-reaction (DFKZ cohort) in univariate analyses (Figure 3), but not in the multivariate analyses, which included disease stage and the Gleason score.

### 3.3. TWIK Expression Signature Displays Differential Oncogenic Signalling Pathway Enrichment

GSEA showed differential enrichment of oncogenic signalling pathways between the TWIK-low and TWIK-high PCa subsets. While the TWIK-high subset showed differential enrichment of oncogenic signalling pathways, including E2F targets, G2-M checkpoint, MYC targets, mitotic spindle, TNF-alpha signalling via NF-kB, KRAS signalling up, mTORC1 signalling, IL-2/STAT5 signalling, inflammatory response, etc. (Appendix A, Figure 4), the TWIK-low subset exhibited differential enrichment in pathways such as oestrogen response early, apical junction, and KRAS signalling down, (Appendix A, Figure 5A). Gene ontology enrichment analysis showed that the TWIK-high subset was differentially enriched for ontology terms such as positive regulation of cell cycle process (GO:0090068), microtubule cytoskeleton organisation involved in mitosis (GO:1902850), DNA metabolic process (GO:0006259), spindle assembly checkpoint signalling (GO:0071173), mitotic spindle assembly checkpoint signalling (GO:0007094), mitotic spindle checkpoint signalling (GO:0071174), negative regulation of mitotic metaphase/anaphase transition (GO:0045841), mitotic cytokinesis (GO:0000281), mitotic cell cycle phase transition (GO:0044772), mitotic spindle organisation (GO:0007052) (Appendix A, Figure 4C), while the TWIK-low subset exhibited enrichment of ontology terms such as muscle contraction (GO:0006936), apical protein localisation (GO:0045176), smooth muscle contraction (GO:0006939), supramolecular fibre organisation (GO:0097435), regulation of aerobic respiration (GO:1903715), xenobiotic catabolic process (GO:0042178), intermediate filament organisation (GO:0045109), actomyosin structure organisation (GO:0031032), and regulation of cell–cell adhesion (GO:0022407) (Appendix A, Figure 5C). The GSEA results indicate that differential TWIK complex expression was associated with differential oncogenic signalling in PCa, thereby validating the biological status of the TWIK complex in prostate carcinogenesis. Furthermore, the GOEA results suggest that TWIK-high was associated with biological processes that support cell proliferation, while the TWIK-low subset was more aligned with biological processes of cell migration.

### 3.4. TWIK Complex Expression Displays Differential Metabolic Pathway Enrichment

The Human-GEM gene set collections were applied to GSEA to demonstrate differential metabolic pathway enrichment between the TWIK complex subsets. The results reveal that while the TWIK-low subset exhibited differential enrichment of the metabolic pathways (KEGG 2021 Human) for valine, leucine and isoleucine degradation, beta-alanine metabolism, propanoate metabolism, arginine and proline metabolism, fatty acid degradation, pyruvate metabolism, histidine metabolism, glycolysis/gluconeogenesis, etc. (Appendix A, Figure 6), the TWIK-high subset showed differential enrichment of ubiquitin-mediated proteolysis, bile secretion, retinol metabolism, ferroptosis, porphyrin and chlorophyll metabolism, carbohydrate digestion and absorption, glycosaminoglycan biosynthesis, etc. (Appendix A, Figure 6). GOEA (GO Molecular Function 2023) confirmed the enrichment of terms such as glutathione transferase activity (GO:0004364), acid-thiol ligase activity (GO:0016878), glutathione binding (GO:0043295), CoA-ligase activity (GO:0016405), aldehyde dehydrogenase (NAD+) activity (GO:0004029), aldehyde dehydrogenase [NAD(P)+] activity (GO:0004030), quaternary ammonium group transmembrane transporter activity (GO:0015651), flavin adenine dinucleotide binding (GO:0050660), succinate transmembrane transporter activity (GO:0015141), etc., in the TWIK-low subset (Appendix A, Figure 6). On the other hand, GOEA showed significant enrichment in ubiquitin-conjugating enzyme activity (GO:0061631), ubiquitin-like protein-conjugating enzyme activity (GO:0061650), acyltransferase activity, transferring groups other than amino-acyl groups (GO:0016747), ubiquitin-like protein transferase activity (GO:0019787), UDP-glycosyltransferase activity (GO:0008194), ubiquitin–protein transferase activity (GO:0004842), heme binding (GO:0020037), monocarboxylic acid binding (GO:0033293), glucuronosyltransferase activity (GO:0015020), etc., among others in the TWIK-high subset (Appendix A, Figure 6). Notably, the TWIK-high PCa subset exhibited enrichment of metabolite terms that specify ubiquitination and glucuronidation reactions and bilirubin metabolism. The overall results show differential enrichment of ontology terms for metabolism between the two subsets. Specifically, the two PCa subsets exhibited differential types of xenobiotics metabolism. While the TWIK-high subset showed preferential enrichment of ubiquitin, glucuronidation, and bilirubin ontology terms, the TWIK-low subset showed enrichment of glutathione detoxification and diverse metabolism ontology terms.

### 3.5. Differential Drug Set Enrichment Characterises TWIK Complex Expression 

The drug sets for ion channel blockers (DSigDBs) were investigated with GSEA in order to identify potential differences in drug responses between the TWIK-high and TWIK-low subsets of PCa. The results demonstrate differential enrichment of the glibenclamide, ethosuximide, valproic acid, dexverapamil, ampyrone, disopyramide, diltiazem response gene sets in the TWIK-high subset (Appendix A and Figure 7A). The TWIK-low subset, on the other hand, showed differential enrichment of the gene sets for phenytoin, lidocaine, amiodarone, tetraethylammonium, and topiramate response (Appendix A). Furthermore, GSEA demonstrated that the TWIK-high PCa subset was enriched in the gene sets which specify responses to crizotinib, daunorubicin, and isotretinoin. The TWIK-low subset was, however, differentially enriched for docetaxel, 6-thioguanine, dactinomycin, indirubin-3’-monoxime, retinoic acid, idarubicin, and vincristine (Appendix A and Figure 7B). The results indicate that TWIK expression may specify differential responses to ion channel blockers and conventional chemotherapeutic drugs. Furthermore, the enrichment of multiple drug sets may indicate that a single biomarker or biomarker panel (such as the TWIK complex comprising *KCNK1*, *KCNK6*, and *KCNK7*) may predict the response to multiple drugs with varying mechanisms of action.

## 4. Discussion

This study investigated the clinicopathological, molecular, and therapeutic significances of TWIK complex expression in three PCa cohorts, confirming that the TWIK gene complex (*KCNK1, KCNK6,* and *KCNK7*) may contribute to tumour progression via its association with the adverse clinicopathological and biological indices of PCa. No other study has investigated the TWIK subfamily of potassium ion channel genes or proteins as a complex, as far as review of the literature shows. The gene co-expression complex approach adopted in this study closely mimics the biological pattern of ion channel activities, inasmuch as ion channels function as complexes or interactive/molecular networks of co-expressed proteins [46,47]. Whilst this study did not find an independent association of the gene complex with indices of tumour progression, i.e., overall and progression-free survival and biochemical recurrence, on the basis of its established function as a membrane potential stabiliser and its association with adverse prognostic clinicopathological (disease stage and Gleason grading indices) and oncogenic signalling indices, the author posits that the TWIK complex contributes to PCa progression. The association in this study of the TWIK complex with cell proliferation, cell migration, and cell cycle signalling is in alignment with some of the findings observed for members of the KCNK gene family in preclinical and a few clinical studies investigating their activities in cancers. For example, Leithner et al. [64] found that *KCNK3* (TASK-1) was highly expressed in the non-small cell lung cancer cell line A549, in which it promotes cell proliferation and inhibits apoptosis. However, this gene was found to inhibit cell proliferation and glucose metabolism in another lung adenocarcinoma cell line model [41]. Moreover, *KCNK6* overexpression enhances cell proliferation, invasion, and migration in a breast cancer cell line model [42]. The same study found that *KCNK6* was overexpressed in a clinical cohort of breast cancer. Furthermore, *KCNK9* promoted cell proliferation and survival by enhancing their resistance to hypoxia and serum deprivation. Knockdown of *KCNK9* inhibited cell proliferation and promoted cell cycle arrest and cellular senescence in the MDA-MB-231 breast cancer model [43]. In addition, *KCNK9* was found to promote migration and survival in gastric cancer cell lines [21] and was upregulated in an animal model of oral squamous cell carcinoma [45]. Also, *KCNK15* (also known as *KCNK14*) was found to be upregulated in pancreatic cancer cell lines, in which it promotes proliferation and migration. In clinical cancer, *KCNK9* and *KCNK18* (TRESK) were downregulated in advanced oral squamous cell carcinoma, in contrast to the findings of the animal study [45]. In their study of KNCK potassium ion channel subfamily expression in hepatocellular carcinoma, Li et al. [28] found that while *KCNK2*, *KCNK15*, and *KCNK17* were downregulated, *KCNK9* was upregulated, and this pattern of expression correlated with better prognosis in patients. In the present study, high expression of the TWIK complex was found to be enriched in cell proliferation, cell cycle, mitotic, signal transduction, and cell migration signalling, which is in agreement with the findings of the studies described above. The association of TWIK expression with clinicopathological and molecular features and with oncogenic signalling pathways validates the relevance of TWIK in clinical prostate carcinogenesis and tumour progression.

TWIK complex expression also exhibited differential enrichment for metabolism and drug response. The implications of these findings are relevant to cancer progression and cancer therapy. First, the association of high TWIK expression tumours with differential metabolism suggests that targeting the TWIK subfamily of genes in clinical PCa may be an authentic strategy to disrupt metabolism in clinical cancer cells and hence to shut down cellular functions. The targeting of specific metabolic processes has been regarded as a veritable strategy for cancer therapy [65]. The activities of the ubiquitination–proteasome system (UPS) were found to be enriched in the TWIK-high PCa subset, evidence of a high protein turnover rate in that subset [66]. Through the primary function of maintenance of tumour cell homeostasis, the UPS is involved in cell cycle progression, apoptosis, angiogenesis, DNA repair, endocytosis, drug resistance, and cell differentiation [66]. In addition, it has been shown that proteasome-independent ubiquitination is widely operative in cancer [67]. Aberrant ubiquitination is associated with deregulation of the cell cycle, cell invasion and metastasis, cancer metabolism, cancer cell stemness, tumour-promoting inflammation, and apoptosis evasion [67]. The association in this study of TWIK complex expression with cell cycle deregulation, cell proliferation, cancer-specific metabolism, oncogenic signalling pathways, inflammatory response, and ubiquitination activities raises the possibility that therapeutic targeting of the TWIK complex may be a potential strategy for targeting more than one hallmark of cancer, inasmuch as ubiquitin and its associated systems have been demonstrated to be druggable in cancer [67]. Furthermore, The TWIK-high PCa subset exhibited enrichment of genes that support glucuronidation reactions and bilirubin metabolism processes. Glucuronidation is a metabolic process that involves the conjugation of drugs and other environmental chemicals with glucuronic acid for the purpose of eliminating these compounds, usually by biliary clearance [68]. The implication of the association of the TWIK complex with enrichment of glucuronidation is that the TWIK-high PCa subset may exhibit high resistance to anti-cancer drugs [68]. However, this finding may also provide opportunity for therapeutic strategies that limit or eliminate drug resistance in the TWIK-high cancer subset [68].

Going further, this study found differential enrichment of ion channel blockers (glibenclamide, ethosuximide, valproic, dexverapamil, ampyrone, disopyramide, and diltiazem) and chemotherapeutic drug sets (crizotinib, daunorubicin, and isotretinoin) in the TWIK-high PCa subset. The association of the TWIK complex expression with multiple drug sets implies that the expression patterns of the TWIK complex may be predictive for multiple drugs with different mechanisms of action, thereby offering the potential for an integrated, multitargeted approach to cancer therapy in PCa management [69]. Using a single biomarker or biomarker panel to predict response to multiple drugs of diverse action mechanisms would increase the therapeutic options available for the cancer patient. It would also permit the deployment of combination chemotherapy without violating the tenets of precision medicine [69]. Combination chemotherapy has the tendency to reduce toxicity in patients, and at the same time, reduce the risks of drug resistance [69]. 

## 5. Conclusions

In conclusion, this study has demonstrated a role of the TWIK interactive network in prostate carcinogenesis and tumour progression, on the bases of its association with the adverse prognostic clinicopathological features of PCa and with known oncogenic signalling pathways. Furthermore, this study has shown that the TWIK complex expression can potentially predict response to multiple drugs and drug treatment strategies, including the targeting of multiple onco-metabolic pathways.

## Figures and Tables

**Figure 1 biology-14-00569-f001:**
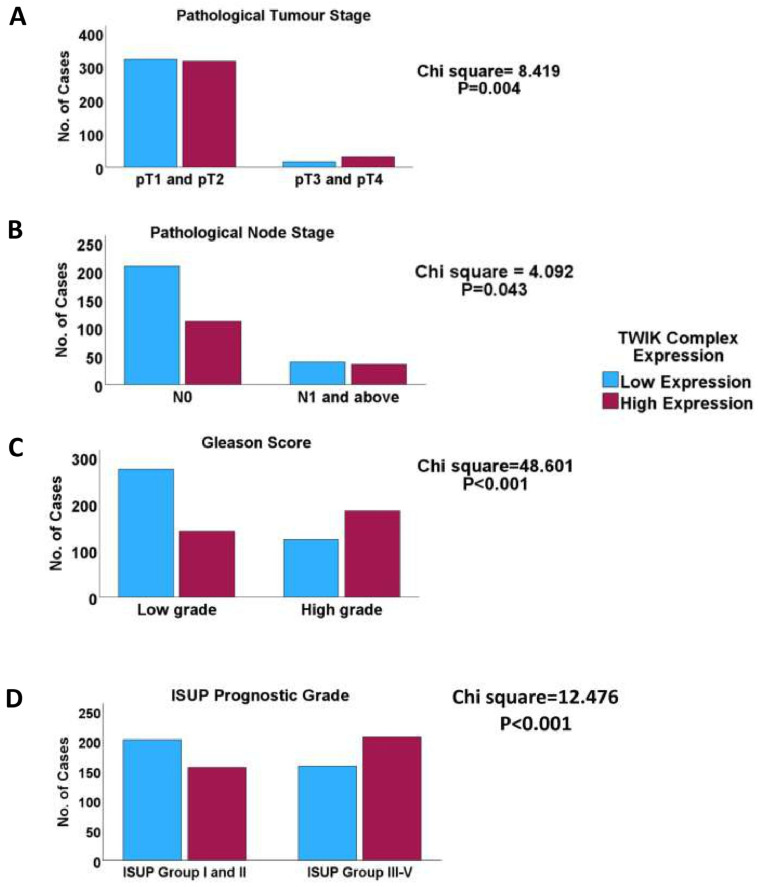
Clustered bar charts showing that high TWIK complex expression is associated with (**A**) the late pathological tumour stage; (**B**) high pathological node stage; (**C**) high Gleason grade, and (**D**) high ISUP prognostic grade groups, evidence that the TWIK ion channels play biological roles in clinical PCa.

**Figure 2 biology-14-00569-f002:**
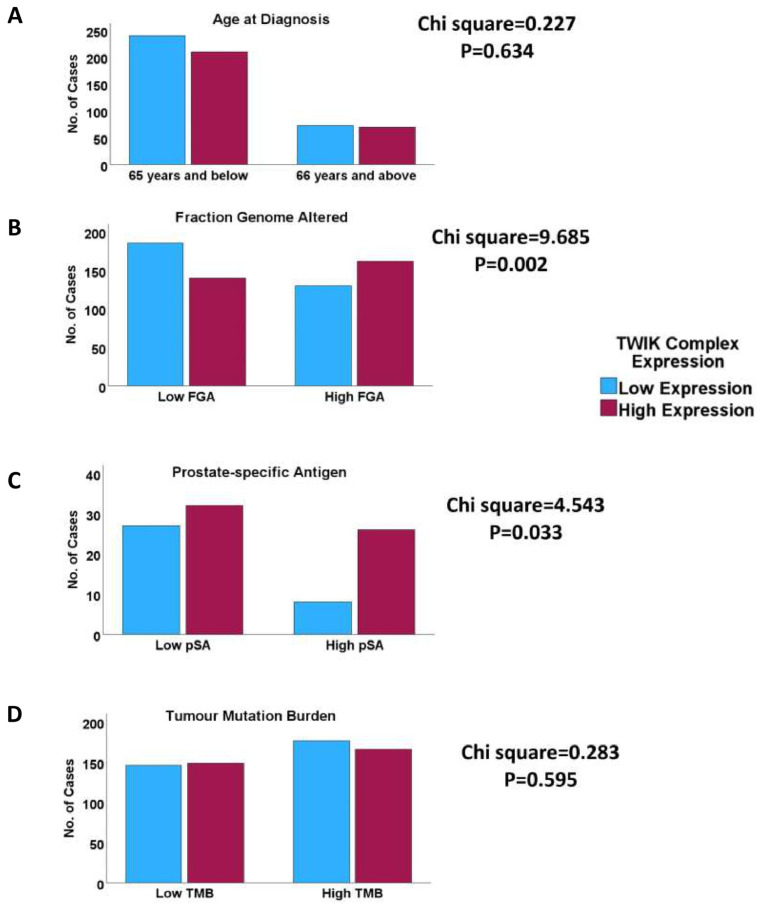
Clustered bar charts comparing TWIK complex expression with clinicopathological, molecular, and biochemical characteristics of clinical PCa. Whilst no association was found for age at diagnosis (**A**) or tumour mutation burden (**D**), high TWIK complex expression was associated with high fraction genome altered (**B**) and high prostate-specific antigen (**C**), evidence that TWIK complex expression may be more relevant to some aspects of clinical PCa biology than others.

**Figure 3 biology-14-00569-f003:**
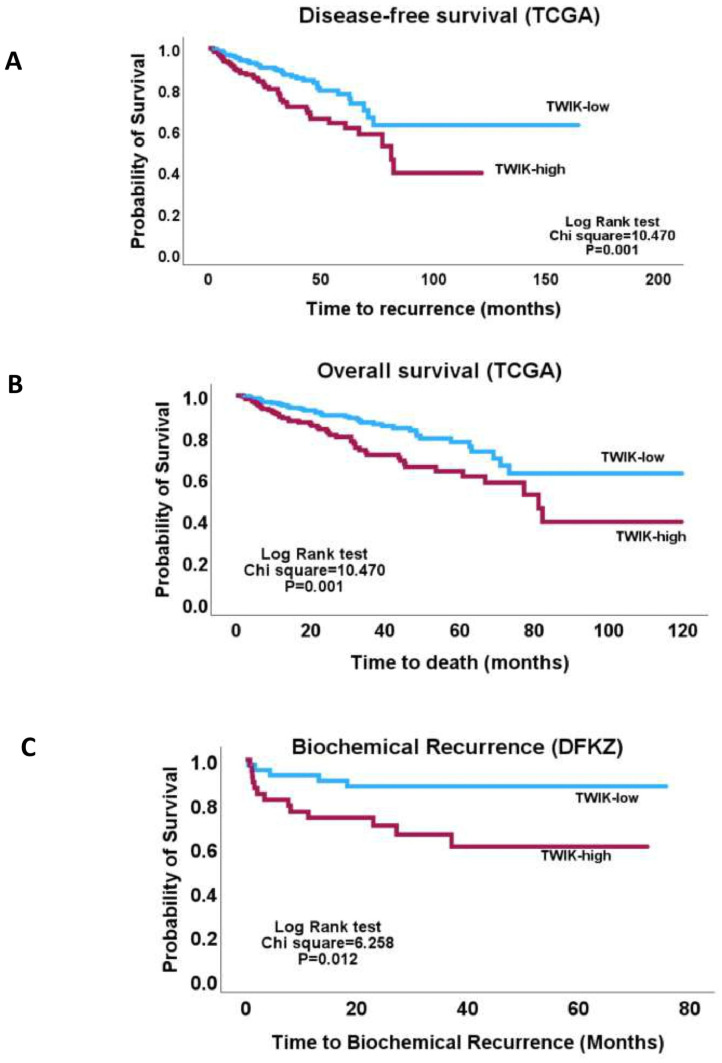
Kaplan–Meier plots showing (**A**) lower disease-free survival in the TWIK-high subset of PCa relative to the TWIK-low subset in the TCGA cohort, (**B**) lower probability of overall survival in the TWIK-high PCa relative to the TWIK-low subset, also in the TCGA cohort, (**C**) lower time-to-biochemical-recurrence in the TWIK-high PCa subset compared to the TWIK-low subset in the DFKZ PCa cohort. Overall, these results suggest that high TWIK complex expression may contribute to PCA tumour biology.

**Figure 4 biology-14-00569-f004:**
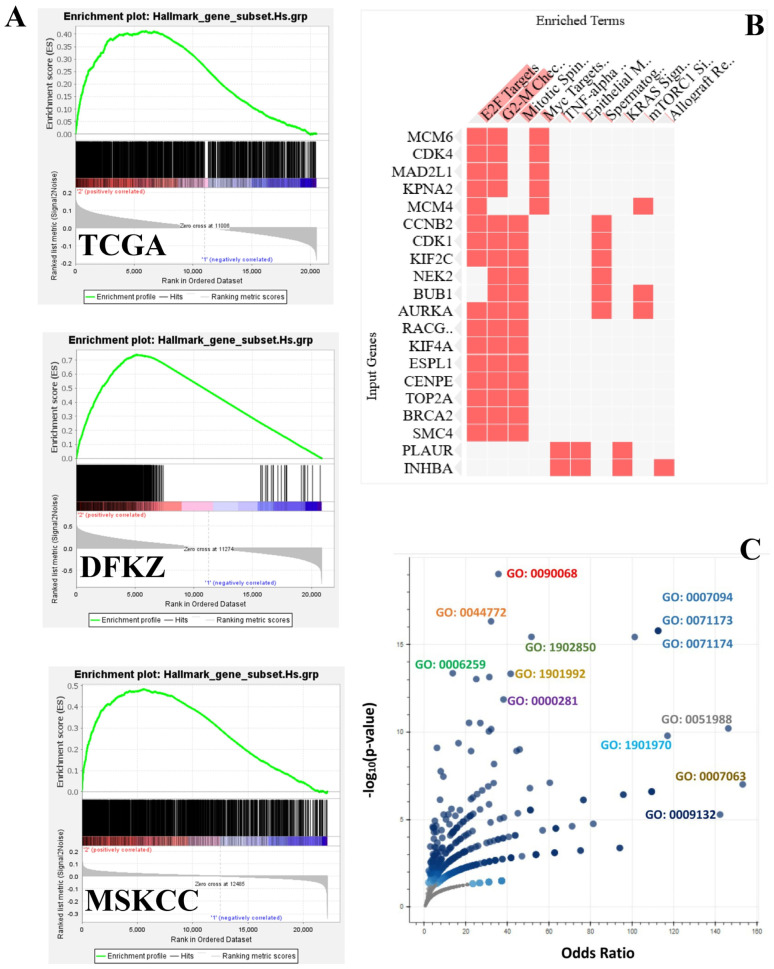
(**A**) Enrichment plots of hallmark gene sets in the TWIK-high PCa subset for the TCGA, DFKZ, and MSKCC cohorts displaying enrichment in cell cycle and proliferation-associated pathways. (**B**) Clustergram showing individual enriched genes that contribute to the enriched cell cycle and cell proliferation pathways. (**C**) Volcano plot showing enrichment of multiple gene ontology terms associated with cell cycle and cell proliferation pathways in the TWIK-high subset, derived from the core enrichment genes shared by all three PCa cohorts. Annotated GO terms explained in the text.

**Figure 5 biology-14-00569-f005:**
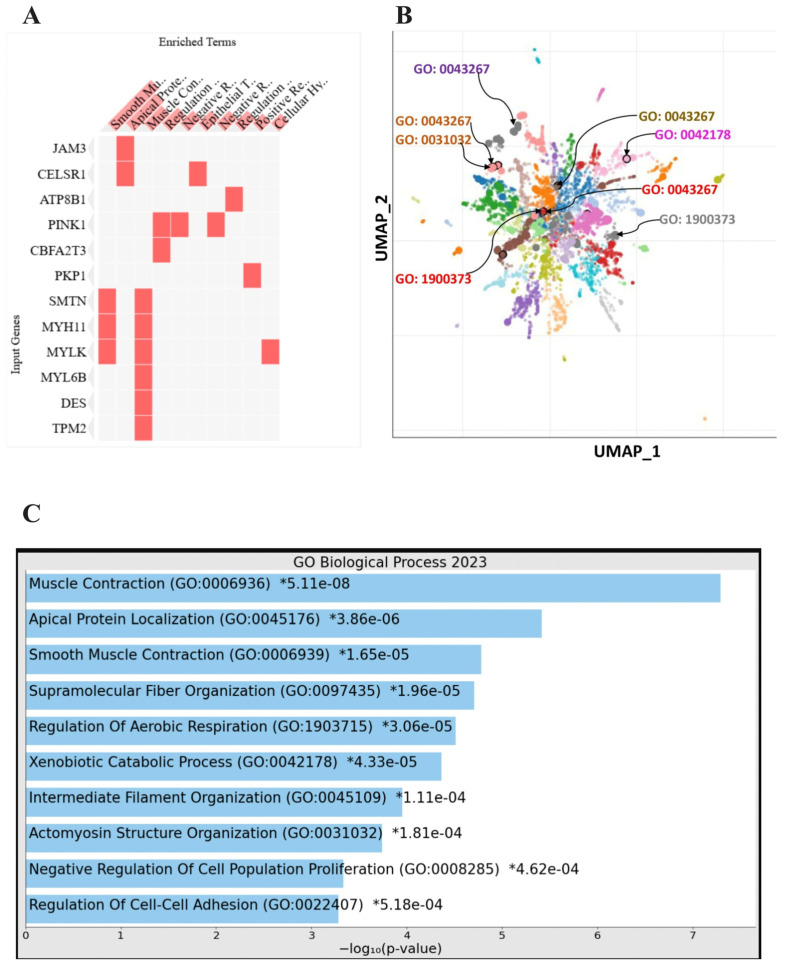
(**A**) Clustergram showing the individual genes that contributed to the enriched pathways in the TWIK-low PCa subset. (**B**) Clustered scatterplot showing the enrichment ontology terms associated with cell migration in the TWIK-low PCa subset; the larger and darker the clusters, the more significantly enriched the ontology terms. (**C**) Bar chart showing the enrichment of ontology terms associated with cell migration in the TWIK-low PCa subset.

**Figure 6 biology-14-00569-f006:**
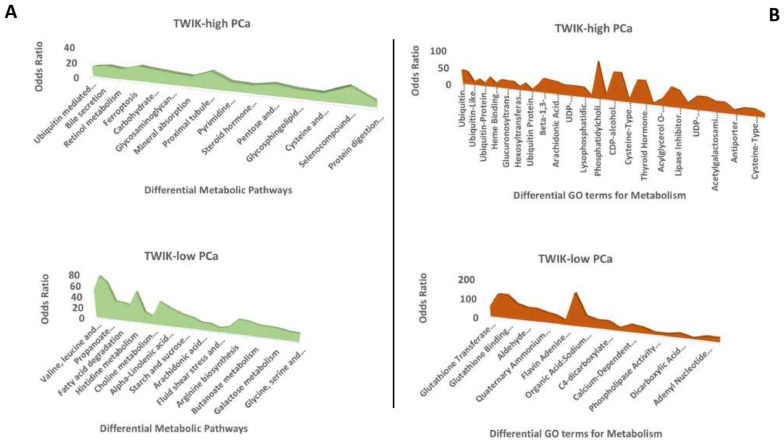
Overlaid area charts of difference showing differential enrichment of (**A**) metabolic pathways and (**B**) metabolite ontology terms between the TWIK-high and TWIK-low subsets of PCa.

**Figure 7 biology-14-00569-f007:**
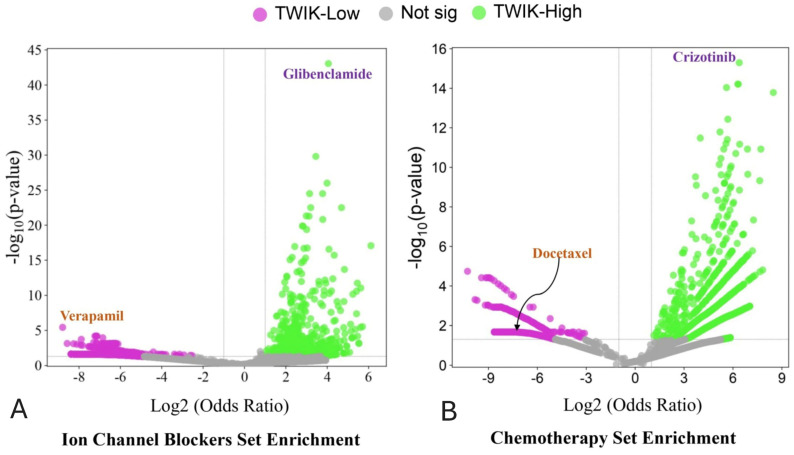
(**A**) Volcano plot of ion channel blockers ontology terms showing differential enrichment of ion channel blockers sets between the TWIK-high and TWIK-low PCa subsets, and (**B**) Volcano plot of chemotherapeutic drug ontology set showing differential enrichment of chemotherapy drug sets between the TWIK-high and TWIK-low PCa subsets. Only a few drugs are annotated here. Not sig = not significant.

## Data Availability

All the genomic and clinicopathological data utilised for this study are freely available in the cBioPortal for Cancer Genomics website (https://www.cbioportal.org/), and the Genome Data Commons repository (https://portal.gdc.cancer.gov/).

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
