# Peer review of "TWIK Complex Expression in Prostate Cancer: Insights into the Biological and Therapeutic Significances of Potassium Ion Channels in Clinical Cancer"

_biology, 2025, doi:10.3390/biology14050569_

Round 1
Reviewer 1 Report (Previous Reviewer 1)
Comments and Suggestions for Authors
Thank you for sending me the updated version of the paper. I have reviewed the improvements made and would like to provide some feedback.
I can confirm that progress has been made compared to the previous version, particularly in terms of the completeness of citations and the clarity of the figure captions. However, there are still some areas that could benefit from further revisions, especially regarding statistical analysis, organization of material and methods.
Author Response
Comment: Thank you for sending me the updated version of the paper. I have reviewed the improvements made and would like to provide some feedback.
I can confirm that progress has been made compared to the previous version, particularly in terms of the completeness of citations and the clarity of the figure captions. However, there are still some areas that could benefit from further revisions, especially regarding statistical analysis, organization of material and methods.
Response: I welcome specific guidance on areas that may require reorganization or expansion. I would be grateful if the reviewer could point out particular sections that need further refinement, and I will be happy to address them accordingly. The methodologies applied in this manuscript are standard statistical and bioinformatics techniques. I would be grateful for specific guidance on which statistical references the reviewer suggests adding to further strengthen the manuscript. I would also appreciate it if the reviewer pointed me in the direction of the error in methodology used in the manuscript.
Reviewer 2 Report (Previous Reviewer 2)
Comments and Suggestions for Authors
The present form of the manuscript reviewed has been sufficiently improved for the publication
Author Response
Comment: The present form of the manuscript reviewed has been sufficiently improved for the publication
Response: I thank this reviewer for his prompt and excellent review of the manuscript
Reviewer 3 Report (New Reviewer)
Comments and Suggestions for Authors
This manuscript is poorly written and lacks scientific rigor. The author performed a few basic bioinformatics analyses on genes related to the TWIK complex and concluded that these genes may be potential therapeutic targets for prostate cancer. However, such conclusions are premature without any functional validation. At a minimum, the author should conduct experimental validation using several prostate cancer cell lines to support the bioinformatics findings. Furthermore, the author appears to have limited familiarity with bioinformatics analysis, as several interpretations are incorrect, and a number of data visualizations are presented in non-standard and difficult-to-interpret formats. For example, Figure 1 should be revised to a dot plot showing individual gene expression levels, grouped by disease stage or pathological grade/score. A similar format should be applied to Figure 2. The GSEA results for DFKZ appear unusual and should be carefully re-examined. Additionally, the GSEA analysis does not provide any evidence supporting the enrichment of cell proliferation pathways (which are suggested by the figure legend). Figures 4a and 4b are difficult to interpret and do not appear to follow standard conventions for pathway enrichment analysis. Figure 4c is mislabeled as a volcano plot; it is unclear what type of plot it represents, as there is no explanation in the figure legend or the results section. Finally, Figure 5b seems to show a UMAP projection commonly used in single-cell RNA-seq data analysis, but this is neither described in the results nor correctly labeled in the legend (which inaccurately refers to it as a scatterplot).
Author Response
Comment 1: This manuscript is poorly written and lacks scientific rigour. The author performed a few basic bioinformatics analyses on genes related to the TWIK complex and concluded that these genes may be potential therapeutic targets for prostate cancer. However, such conclusions are premature without any functional validation. At a minimum, the author should conduct experimental validation using several prostate cancer cell lines to support the bioinformatics findings.
Response 1: Thank you for raising this point. The study focuses on the clinicopathological, biological, and therapeutic implications of KCNK2, KCNK9, KCNK15, and KCNK17 as a functional complex in clinical prostate cancer samples. These analyses were conducted using over 800 clinical cases from TCGA and cBioPortal databases, reflecting gene expression in the natural tumour microenvironment.
While experimental validation using prostate cancer cell lines can provide additional mechanistic insights, it is important to note that clinical datasets offer a more biologically relevant context for several reasons. First, bulk tumour samples encompass both cancer cells and their surrounding tumour microenvironment, which is absent in traditional in vitro systems. Second, multiple studies have demonstrated that cell cultures do not always recapitulate the biology of natural cancers, particularly in terms of gene expression profiles, signalling dynamics, and microenvironmental interactions (doi: 10.1242/dmm.037366; doi: 10.3389/fonc.2021.782766; https://doi.org/10.1016/j.bbcan.2018.06.004).
Nevertheless, future mechanistic investigations may consider in vitro models as supportive tools to explore specific signalling pathways or drug response mechanisms—provided their well-recognised limitations in mimicking the complexity of clinical tumour environments are taken into account.
Comment 2: Furthermore, the author appears to have limited familiarity with bioinformatics analysis, as several interpretations are incorrect, and a number of data visualizations are presented in non-standard and difficult-to-interpret formats.
Response 2: Thank you for your feedback. The bioinformatics analyses applied in this study—including gene set enrichment, gene ontology enrichment, and drug set enrichment—are widely accepted, well-documented, and frequently used in cancer transcriptomics research. The data visualizations were primarily generated using platforms such as Enrichr and GSEA, which produced figures in standard formats recognized by the bioinformatics community. Nonetheless, I acknowledge that the interpretation of such plots can vary depending on the reader’s familiarity with these tools. To enhance clarity, I have added annotations to the figures and expanded the figure legends. If there are specific aspects that appear incorrect or unclear, I would be grateful for guidance so they can be addressed in future work.
Comment 3: For example, Figure 1 should be revised to a dot plot showing individual gene expression levels grouped by disease stage or pathological grade/score. A similar format should be applied to Figure 2.
Response 3: Thank you for the suggestion. Figures 1 and 2 were generated from statistical analyses conducted in SPSS, and the analyses focused on the TWIK complex as a functional unit rather than on individual gene behaviour. This approach reflects the biological premise that the KCNK2, KCNK9, KCNK15, and KCNK17 genes operate together as an ion channel complex in prostate cancer. The figures display associations between the overall complex expression and clinicopathological features using one-way ANOVA and chi-square tests, which are commonly employed for such categorical comparisons. These results are further supported by detailed outputs provided in the Supplementary Materials. I acknowledge that dot plots per individual gene may offer additional granularity, but given the biological hypothesis and complex-level focus of the current study, the existing format was deemed more appropriate for the questions addressed. Future work may expand this analysis to include gene-level comparisons where relevant. This rationale has now been explicitly clarified in the revised manuscript.
Comment 4: The GSEA results for DFKZ appear unusual and should be carefully re-examined. Additionally, the GSEA analysis does not provide any evidence supporting the enrichment of cell proliferation pathways (which are suggested by the figure legend).
Response 4: Thank you for the comment. The GSEA results for the DFKZ cohort were generated using the Hallmark Cell Proliferation gene set and follow standard output conventions. Although the enrichment score and plot format may vary depending on the dataset size and gene ranking metrics, the visualisation reflects the actual enrichment outcome. To clarify the association with cell proliferation, the gene ontology enrichment analysis—detailed in both the main text and Supplementary Materials—demonstrated significant enrichment for multiple proliferation-related pathways in the TWIK-high subset, including “Positive Regulation of Cell Cycle Process” (GO:0090068), “DNA Metabolic Process” (GO:0006259), and “Mitotic Spindle Organisation” (GO:0007052), among others. To avoid misinterpretation, I have revised the figure legend and narrative to better explain the biological significance of these enrichment patterns.
Comment 5: Figures 4a and 4b are difficult to interpret and do not appear to follow standard conventions for pathway enrichment analysis. Figure 4c is mislabeled as a volcano plot; it is unclear what type of plot it represents, as there is no explanation in the figure legend or the results section. Finally, Figure 5b seems to show a UMAP projection commonly used in single-cell RNA-seq data analysis, but this is neither described in the results nor correctly labelled in the legend (which inaccurately refers to it as a scatterplot).
Response 5: The bar charts, clustergrams, volcano plots, and scatterplots were generated using established bioinformatics platforms such as Enrichr and MDBSig GSEA. These figures were exported in their native formats and compiled into the manuscript without modification. It is understood that visual conventions may vary across enrichment analysis tools, which can sometimes result in figures appearing unfamiliar to readers who are more accustomed to other platforms. Based on reviewer comments, I recognized the potential for such differences in perception and have therefore included additional annotations in selected figures, particularly the volcano and scatterplots, to enhance clarity. There is no universal convention for presenting gene set, gene ontology, or pathway enrichment results. The various tools available to researchers (e.g., Enrichr, DAVID, GSEA) each generate visual outputs in their own style. Researchers tend to become familiar with specific tools, and variations between platforms are to be expected. Specifically, Figure 4A presents enrichment plots from MDBSig GSEA, which has maintained a consistent plot structure since its initial development. Figure 4B is a clustergram from Enrichr that illustrates individual gene enrichment within cancer-related pathways. Figure 4C, although different from classical volcano plots, follows Enrichr’s internal format and is accurately labelled according to that system. Likewise, Figure 5B is a scatterplot produced by Enrichr, displaying both significant and non-significant enrichment terms in the TWIK-high prostate cancer subset. In this plot, dot colour represents functional pathway clusters, and dot size reflects the correlation magnitude with TWIK expression. With regard to the biological interpretation, I have already described in the manuscript that the TWIK-low subset exhibited enrichment for ontology terms such as “Muscle Contraction”, “Apical Protein Localization”, and “Regulation of Cell-Cell Adhesion”, while the TWIK-high subset was enriched in proliferation-associated processes such as “Positive Regulation of Cell Cycle” and “Mitotic Spindle Organisation” (Supplementary Materials 2, Figure 5). These findings support the association between TWIK complex expression and differential oncogenic signalling in prostate cancer and thus validate the biological relevance of the TWIK complex.
The following modifications were made to improve clarity:
1. Annotations were added to Figures 4C and 5B to support interpretation of the enrichment outputs.
2. The volcano plot in Figure 4C was annotated with key ontology terms showing the highest –log10(p-values), and further detail is available in the Supplementary Materials.
3. Enhancement of the font sizes of the clustergrams to clarify the data they display
Reviewer 4 Report (New Reviewer)
Comments and Suggestions for Authors In the current bioinformatics study, Dr. Alfahed analyzed the association of TWIK subfamily of potassium channels, namely KCNK1, 6, and 7, with key clinicopathological characteristics of prostate cancer based on data deposited in TCGA and cBioPortal database. Comparing samples from patients with high and low TWIK expression, the author showed associations of the high TWIK group with pathological node stage for the N0 group, Gleason score in dependence of cancer grade, prostate-specific antigen expression, as well as disease-free and overall survival and time to biochemical recurrence. Functional analysis performed by Dr. Alfahed in the second part of the study showed that high TWIK expression was associated with processes related to cell proliferation and ubiquitin degradation system, whereas low TWIK expression was associated with cell motility and glutathione metabolism. As expected, high TWIK expression was also enriched for ion channel blockers in the DSigDB database. Based on these findings, Dr. Alfahed argues for the involvement of the TWIK system in the pathogenesis of prostate cancer and supports this hypothesis in the Discussion with published data on other cancers. In my opinion, this study can be published in Biology after some corrections. (1) Figures 4B, 5A - In my opinion, the results of functional analysis should be presented in the form of bar graphs with the names of the functional terms on the OY axis and -log10(p-value) on the OX axis. Then it will be possible to understand which processes are most associated with the identified differentially expressed genes. Please correct. Although the heat map gives information about the genes involved in a particular pathway/process, it is not very informative. (2) Figure 4C - It is not clear from the text of the article what is shown in this figure. Please provide more information. (3) Figure 5B - It is not clear what is shown in the UMAP plot. What are these clusters? How does this relate to the results described in the manuscript? Please provide a detailed description. (4) Section 3.5 - Dear Dr. Alfahed, in order to better understand your results, you should generate a figure that includes information on drug enrichment. (5) Text edits: p. 2, end of second paragraph - please insert [ before 38. p. 12, 4th line from bottom - in my opinion, "methods for xenobiotic metabolism" is an unfortunate expression for cellular processes. Better to replace with more appropriate synonyms, e.g., "Specifically, the two PCa subsets exhibited different types of xenobiotic metabolism." p. 14, end of first paragraph - in the sentence "Li et al [28] found that while KCNK2, KCNK9, <...> were downregulated, KCNK9 was upregulated" is an error. Up- or down-regulation for KCNK9 is shown by Li et al? p. 15, line 5 - typo in the phrase "cancer sell". Please delete. p. 15, Conclusions, line 3 - extra with. Please delete.Author Response
Comment 1: In the current bioinformatics study, Dr. Alfahed analyzed the association of the TWIK subfamily of potassium channels, namely KCNK1, 6, and 7, with key clinicopathological characteristics of prostate cancer based on data deposited in the TCGA and cBioPortal databases. Comparing samples from patients with high and low TWIK expression, the author showed associations of the high TWIK group with pathological node stage for the N0 group, Gleason score in dependence of cancer grade, prostate-specific antigen expression, as well as disease-free and overall survival and time to biochemical recurrence. Functional analysis performed by Dr. Alfahed in the second part of the study showed that high TWIK expression was associated with processes related to cell proliferation and the ubiquitin degradation system, whereas low TWIK expression was associated with cell motility and glutathione metabolism. As expected, high TWIK expression was also enriched for ion channel blockers in the DSigDB database. Based on these findings, Dr. Alfahed argues for the involvement of the TWIK system in the pathogenesis of prostate cancer and supports this hypothesis in the Discussion with published data on other cancers. In my opinion, this study can be published in Biology after some corrections.
Response 1: I thank this reviewer for his excellent and thorough review of the manuscript.
The bar charts, clustergrams, volcano plots, and scatterplots were produced online by the gene ontology enrichment analyses in the Enrichr environment and were copied as is and compiled into the figures in the manuscript. Based on reviewers’ comments, I realise that the reader may not see the figures as they may be in the author’s mind’s eye. Therefore, I have included annotations in some of the figures, especially in the volcano and scatterplots.
Comment 2: (1) Figures 4B and 5A— In my opinion, the results of functional analysis should be presented in the form of bar graphs with the names of the functional terms on the OY axis and -log10(p-value) on the OX axis. Then it will be possible to understand which processes are most associated with the identified differentially expressed genes. Please correct. Although the heat map gives information about the genes involved in a particular pathway/process, it is not very informative.
Response 2: Thank you for this valuable suggestion. I acknowledge the importance of bar charts for clearly displaying the significance levels of functional pathways. However, I retained the clustergrams in Figures 4B and 5A because they provide complementary information through highlighting the individual genes involved in each enriched pathway. To enhance clarity, I have now increased the font sizes of the displayed genes and pathways and clarified their purpose in the figure legends. Additionally, the full list of enriched pathways and associated p-values is provided in the Supplementary Materials. This combined approach aims to offer a broader and more informative view of the enrichment results.
Comment 3: Figure 4C - It is not clear from the text of the article what is shown in this figure. Please provide more information.
Response 3: Thank you for pointing this out. I have updated Figure 4C by annotating the volcano plot to highlight the most significantly enriched functional pathways in the TWIK-high PCa subset. The annotation focuses on those pathways with the highest -log10(p-values), allowing readers to quickly grasp the key biological processes involved. Furthermore, the Supplementary Materials include the full enrichment results for reference. I also revised the figure legend and relevant section in the manuscript to clarify what is being shown in this plot.
Comment 4: Figure 5B - It is not clear what is shown in the UMAP plot. What are these clusters? How does this relate to the results described in the manuscript? Please provide a detailed description.
Response 4: Thank you for this observation. Figure 5B is a scatterplot generated using the Enrichr environment, not a UMAP plot. It displays functional pathways enriched in the TWIK-high PCa subset. Each dot represents a pathway, with the size of the dot indicating the magnitude of correlation (enrichment score) and the colour denoting the cluster to which the pathway belongs. Annotations have now been added to clarify the identity of key pathways and their significance levels. I have also revised the figure legend and manuscript text to avoid confusion and ensure accurate interpretation.
Comment 5: Section 3.5 - Dear Dr. Alfahed, in order to better understand your results, you should generate a figure that includes information on drug enrichment.
Response 5: Thank you for the helpful suggestion. In response, I have generated a new scatterplot using a different Bioinformatics platform to visually present the top drug candidates enriched in the TWIK-high PCa subset. These visualizations highlight the most relevant compounds based on enrichment scores and provide a clearer understanding of the potential therapeutic implications. The new figures have been added to Section 3.5 of the manuscript, and the figure legend has been updated accordingly.
Comment 6: Text edits:
6.1. p. 2, end of second paragraph - please insert [ before 38.
Response 6.1. Done
6.2. p. 12, 4th line from bottom—In my opinion, "methods for xenobiotic metabolism" is an unfortunate expression for cellular processes. Better to replace with more appropriate synonyms, e.g., "Specifically, the two PCa subsets exhibited different types of xenobiotic metabolism."
Response 6.2. Thank you for your suggestion; the sentence has been revised.
6.3. p. 14, end of first paragraph - in the sentence "Li et al. [28] found that while KCNK2, KCNK9, <...> were downregulated, KCNK9 was upregulated" is an error. Up- or down-regulation for KCNK9 is shown by Li et al?
Response 6.3. The sentence has been corrected to read “Li et al. [28] found that while KCNK2, KCNK15, and KCNK17 were downregulated, KCNK9 was upregulated…,” as the cited article shows.
6.4. p. 15, line 5 - typo in the phrase "cancer sell." Please delete.
Response 6.4. Done
6.5. p. 15, Conclusions, line 3 - extra with. Please delete.
Response 6.5. Done
Round 2
Reviewer 4 Report (New Reviewer)
Comments and Suggestions for Authors
I thank Dr. Alfahed for his careful attention to my comments. All comments have been duly corrected or commented.
I would still try to avoid long names on the axes of heatmaps and diagrams that end with an ellipsis (...) due to space constraints. Alternatively, full names of functional terms can be placed vertically and gene names horizontally (Fig. 4A, 5A). Such an arrangement would allow the reader to understand the data more clearly. In the case of Figure 6, I would show the full names of the functional terms and rearrange the figure. However, these comments are only recommendations for future articles - as Dr. Alfahed rightly pointed out, all the information obtained is duplicated in the Supplementary Materials.
In my opinion, the manuscript is ready for publication. I wish Dr. Adbulaziz Alfahed every success in his research!
This manuscript is a resubmission of an earlier submission. The following is a list of the peer review reports and author responses from that submission.
Round 1
Reviewer 1 Report
Comments and Suggestions for Authors The work presented by the authors seems to be quite lacking from many points of view. Starting with the introduction, where there are several inaccuracies, more or less serious, in the description of the molecular complexes under study and the functions they perform, both in a physiological and pathological condition. Furthermore, in the same section, the references provided by the authors do not adequately support the claims made, following some examples:- “Specifically, this study investigated the expression in PCa of the TWIK (Two P-domain in a weakly inward rectifying K+) subfamily of the KCNK channels, a family of the two-pore domain potassium channels, member K”. This description should be rephrased in a more understandable way.
- “Furthermore, it has been demonstrated that targeting ion channels may be a veritable strategy for the treatment of cancer. Therapeutic targeting of ion channels is an attractive treatment strategy [13], since ion channels appear to have ubiquitous functions in tumour cells in vitro. However, these findings have yet to be translated into clinical use; studies validating the clinical usefulness of the findings of the in vitro studies are few.” More References are needed
- “The maintenance of membrane potentials, in turn, is essential for many cellular functions, including metabolic regulations, apoptosis, chemoperception, etc.” More References are needed
The language must be improved
Reviewer 2 Report
Comments and Suggestions for Authors
The author presented data to gain insight into TWIK complex expression in prostate cancer and its biological and therapeutic significances in clinical prostate cancer.
Nevertheless, several issues about this manuscript need to be considered:
-in general the manuscript requires an extensive revision, in all sections: introduction, methods, results and discussion.
-data presented in fig.2 clearly indicate that there is not a strictly correlation among low and high TWIK expression and the different parameters considered. This observation could be considered in the discussion.
-data in figure 4, 5, and 6 are not clearly presented: there is not a appropriate caption to describe all those figures. The related results also need to be improved in the description
-Moreover, the discussion did not provide an exhaustive explanation of data obtained. The data need to be discussed in a more detailed way.
Comments on the Quality of English Language
The English could be improved to more clearly express the research
